# Comparison of Postoperative Stability of Intraocular Lenses after Phacovitrectomy for Rhegmatogenous Retinal Detachment

**DOI:** 10.3390/jcm11123438

**Published:** 2022-06-15

**Authors:** Ayaka Akiyama, Harumasa Yokota, Hiroshi Aso, Hirotsugu Hanazaki, Masanori Iwasaki, Satoru Yamagami, Taiji Nagaoka

**Affiliations:** 1Division of Ophthalmology, Department of Visual Sciences, Nihon University School of Medicine, Tokyo 156-0052, Japan; akiyama.ayaka@nihon-u.ac.jp (A.A.); asox555@gmail.com (H.A.); hanazaki.hirotsugu@nihon-u.ac.jp (H.H.); iwasakicom@yahoo.co.jp (M.I.); yamagami.satoru@nihon-u.ac.jp (S.Y.); taijinagaoka@gmail.com (T.N.); 2Aso Eye Clinic Kyodo, Tokyo 156-0052, Japan

**Keywords:** intraocular lens, gas tamponade, phacovitrectomy, retinal detachment, single-piece intraocular lens, three-piece intraocular lens

## Abstract

We retrospectively compared the stability of intraocular lenses (IOLs) routinely used at our institution by measuring IOL position after phacovitrectomy for rhegmatogenous retinal detachment (RRD). Patients with RRD who underwent phacovitrectomy with gas tamponade received one of three IOLs: 6-mm, single-piece NS-60YG (NIDEK, 15 eyes); 6-mm, single-piece XY1 (HOYA, 11 eyes); or 7-mm, three-piece X-70 (Santen, 11 eyes). Various parameters associated with the anterior chamber, lens, and IOL were measured by swept-source anterior segment optical coherence tomography (CASIA2; Tomey Corp) before and 1 week and 1 month after surgery. IOL position was determined as follows: IOL position = (postoperative aqueous depth [AQD] − preoperative AQD)/lens thickness. We found no significant difference in axial length between the IOLs (*p* = 0.97). At 1 week, IOL position was as follows: NS-60YG, 0.32; XY1, 0.24; and X-70, 0.26 (*p* < 0.05). The respective IOL positions at 1 month were 0.35, 0.27, and 0.28 (*p* < 0.01). These results indicated the smallest anterior shift with NS-60YG. To replicate the anterior shift of IOL position ex vivo, biomechanical measurement was performed. NS-60YG resisted more displacement force than the other IOLs. Thus, in eyes undergoing phacovitrectomy for RRD, NS-60YG was the most stable of the three IOLs studied.

## 1. Introduction

Rhegmatogenous retinal detachment (RRD) is an acute retinal disorder that typically requires immediate treatment. Scleral buckling and pars plana vitrectomy (PPV) are the main surgical treatments, and these procedures obtain a high rate of successful reattachment with a single operation [1]. Scleral buckling is generally used in younger patients, and PPV is more likely to be performed in middle-aged patients. To avoid the need for subsequent cataract surgery and thus reduce the burden on patients, RRD can be treated by phacovitrectomy [2,3,4], which achieves retina reattachment using a gas or silicone tamponade. Although phacovitrectomy has a reasonable success rate, complications related to IOL stability may occur, such as myopic shift, IOL iris capture, and IOL decentration [3]. Among these complications, myopic shift occurs the most often and remains even after the tamponade materials have disappeared [2,5].

Myopic shift has been attributed to several factors, such as the simple replacement of vitreous with aqueous humor, gas tamponade, and IOL type [6]. Among those factors, IOL type is the easiest to influence. Although single-piece IOLs have become popular for use in cataract surgery [7], three-piece IOLs also have been used to improve visibility during phacovitrectomy because of their larger diameter (7 mm). However, no study has compared the stability of single- and three-piece IOLs.

Recently, Shiraki et al. determined the exact IOL position after phacovitrectomy with a new method, swept-source anterior segment optical coherence tomography (SS-ASOCT) [2,8]. They concluded that gas tamponade-induced anterior shift of IOLs resulted in myopic error values of approximately 0.5 diopters (D). Furthermore, they suggested that IOL position determined by SS-ASOCT parameters is useful for longitudinally observing postoperative IOL position [2].

In our hospital, there was a period when surgeons used mainly three different IOLs in phacovitrectomy for RRD. By retrospectively reviewing patient charts, the present study investigated the postoperative stability of these three IOLs. In addition, biomechanical measurements were performed to characterize the IOLs used in our hospital.

## 2. Materials and Methods

### 2.1. Study Design and Participants

This study was a retrospective, observational study. Individuals were eligible for inclusion if they underwent phacovitrectomy (25 or 27 gauge) for primary RRD and gas tamponade with 20% sulfur hexafluoride gas (SF_6_) at Nihon University Itabashi Hospital between May 2020 and March 2021. The decision to use a 25 or 27 gauge was made by the surgeon. To minimize inter-individual variation, we used the following exclusion criteria: (1) intraoperative posterior capsule rupture; (2) corneal abnormalities, such as keratoconus; (3) intraocular tamponade with silicone oil or octafluoropropane (C_3_F_8_); (4) scleral buckle complex vitrectomy; and (5) lack of preoperative testing with CASIA2 (Tomey Corp., Nagoya, Japan). All participants were grouped according to the implanted IOL.

The study protocol was approved by the ethics committee of Nihon University (RK-200114-11), and all procedures were implemented in accordance with the tenets of the Declaration of Helsinki. All participants provided written informed consent.

### 2.2. IOLs

Nidek NS-60YG, HOYA XY1, and Santen X-70 were chosen as the IOLs of interest because they were the main IOLs used for phacovitrectomy in RRD during the study period. The characteristics of the IOLs are summarized in Table 1. NS-60YG and XY1 are single-piece soft acrylic IOLs with a 6.0 mm optic diameter, and X-70 is a three-piece IOL with a 7.0 mm optic diameter and polyvinylidene difluoride haptics.

### 2.3. Operative Procedures

For phacoemulsification, a 2.4 mm clear scleral corneal incision was made, and then 25- or 27-gauge pars plana vitrectomy was performed with the Constellation Vision System (Alcon Laboratories, Inc., Fort Worth, TX, USA) and the Resight Fundus Viewing System (Carl Zeiss Meditec Inc., Dublin, CA, USA). To minimize residual vitreous cortex, the surgeon performed core vitrectomy, midperipheral vitrectomy, and vitreous base shaving under the scleral depression. Sometimes, the surgeon used perfluorocarbon liquid (Perfluoron; Alcon Laboratories, Inc., Geneva, Switzerland). The IOL was implanted into the capsule bag. Fluid/air exchange was performed, and then endolaser photocoagulation was applied around the area of the retinal breaks. If there were any signs of leakage after air injection, the sites were sutured to maintain an air-tight condition. Then, the vitreous cavity was filled with 20% sulfur hexafluoride. Patients were instructed to remain in a prone position for a maximum of three days.

### 2.4. Assessments

Before phacovitrectomy, the anterior segment parameters were measured and defined with SS-ASOCT, and the anterior chamber width (ACW), aqueous depth (AQD), central corneal thickness (CCT), and lens thickness (LT) were defined and automatically measured with CASIA2 (software version 3E.26). Axial lengths were measured by optical biometry (IOL Master; Carl Zeiss) in the case of macula-on RRD or applanation A-scan ultrasonography (Tomey Corp., Nagoya, Japan) in the case of macula-off RRD.

At 1 week and 1 month after surgery, refraction (expressed as a sphere equivalent) and CASIA2 parameters were measured. The postoperative AQD was defined as the distance between the anterior IOL surface and the posterior corneal surface and was measured and calculated automatically with CASIA2, as were the tilt and decentration of the IOL. The IOL position was expressed as the ratio of the difference between the preoperative and postoperative AQD to the thickness of the lens, as previously reported [2], i.e., as (postoperative AQD–preoperative AQD) to lens thickness (Figure 1).

The IOL displacement force was measured as described elsewhere [7]; however, we modified the measurement because the three-piece IOL acts differently from the single-piece IOLs [9]. All the selected IOLs were +20 diopter (D). Briefly, the IOLs were placed in a fixture at room temperature (23 °C, Figure 2A). To replicate the force induced by gas tamponade during and after phacovitrectomy, the IOLs were pushed on the posterior side by a digital micrometer head (diameter, 3.0 mm; Mitutoyo, Kawasaki, Japan) until they moved 0.5 mm anteriorly (Figure 2A). The IOL displacement force was measured (in mN) by an electronic balance (SHIMADZU, Kyoto, Japan) and recorded at each 0.1 mm anterior movement until the anterior displacement reached 0.5 mm. In addition, to evaluate the strength of the haptics of the IOLs, we calculated the mean compression load by dividing the total compression load by the contact angle; the total compression load was defined as the force required to decrease the total length of the IOL to 10 mm (Figure 2B). The contact angle of the IOL was measured according to Annex E: Measurement of angle of contact (ISO11979-3:2012). Briefly, an IOL was set in a 10 mm diameter holder and imaged by a profile projector (Mitutoyo), and the contact angle was defined as the total contact angle of the two haptics (Figure 2C).

The following data were collected: preoperative and postoperative best-corrected visual acuity (BCVA), preoperative and postoperative refraction, ACW, AQD, CCT, and LT. The mean refractive prediction error (ME; i.e., the actual postoperative refraction minus the preoperative refraction predicted by the formula for the exact power of the implanted IOL) and the mean median absolute error (MedAE) were calculated at 1 month postoperatively and compared between the groups.

The main outcome measures were the associations between CASIA2 parameters, IOL position, and refractive outcomes.

### 2.5. Statistical Analysis

Continuous values are expressed as mean ± standard deviation. For statistical analyses, the BCVA of each patient was converted to its logarithm of the minimal angle of resolution (logMAR) value. Data were analyzed with the Ekuseru-Toukei 2010 (Social Survey Research Information Co., Ltd., Tokyo, Japan). A *p*-value of less than 0.05 was considered statistically significant.

## 3. Results

### 3.1. Patient Characteristics

Patient characteristics are summarized in Table 2. The study included data from 37 eyes in 36 patients. Among the 37 eyes, the three types of IOL were implanted as follows: NS-60YG, 15 eyes; XY1, 11 eyes; and X-70, 11 eyes. The mean age of all patients was 60.6 ± 8.3 years (range, 50–83 years), and the mean axial length was 25.6 ± 1.4 mm. The mean BCVA (logMAR) was 0.30 ± 0.60 before surgery and improved significantly to −0.02 ± 0.18 at 1 month.

### 3.2. Postoperative Refractive Error

At 1 month postoperatively, the overall MedAE was 0.44 ± 0.33, and the overall ME was −0.39 ± 0.39, indicating an anterior shift of IOL position (Table 3). MedAE and ME were not significantly different between groups at 1 month postoperatively, but there was a trend toward less refractive error in the NS-60YG group than in the XY1 and X-70 groups.

### 3.3. Swept-Source Anterior Segment Optical Coherence Tomography Parameters and Intraocular Lens Position

Postoperative SS-ASOCT parameters are shown in Table 4. AQD showed no difference between the IOLs at 1 week and 1 month after surgery. At 1 week, NS-60YG was significantly closer to the center of the lens than XY1 and X-70 (*p* < 0.05). At 1 month, NS-60YG was still significantly closer to the center of the lens than XY1 and X-70 (*p* < 0.01).

### 3.4. Simple Regression Analysis between ME and Variables

A significant correlation was found only between macular status and ME, and the type of IOL was not significantly correlated with ME (Table 5). Macula-off RRD resulted in a significant myopic error compared with macula-on RRD (Figure 3).

### 3.5. IOL Displacement Force, Contact Angle, and Compression Load

The displacement forces of each IOL are summarized in Figure 4A. NS-60YG required a greater force than XY-1 and X-70 to reach the same anterior displacement. There was also a significant difference in IOL displacement force between XY1 and X-70. Representative images for contact angle measurements of each IOL are shown in Figure 4B. The contact angle of NS-60YG (160°) was greater than that of XY1 (82°) and X70 (78°), and the unit contact load of NS-60YG was the smallest, indicating that the haptics of NS-60YG are capable of evenly attaching the capsule (Figure 4C).

## 4. Discussion

To the best of our knowledge, this is the first study to investigate the anterior shift of various IOLs after phacovitrectomy for RRD and to measure the displacement force of the IOLs. We found a difference in the anterior shift of the three main IOLs (NS-60YG, XY1, and X-70) used during the study period for phacovitrectomy in RRD at our institution. In the eyes with gas tamponade, NS-60YG showed better postoperative stability than XY1 and X-70. The biomechanical values of the displacement forces also indicated good postoperative stability of NS-60YG after phacovitrectomy for RRD.

The choice of IOL depends on a surgeon’s preference. The operations in the current study were performed by three surgeons who mainly used the three IOLs studied here. NS-60YG and XY1 are one-piece IOLs with the same optic diameter and total length, but different haptic designs, and X-70 is a three-piece IOL with a larger optic diameter that yields a wider clear view of the peripheral retina. As expected, IOL selection did not affect postoperative BCVA. However, we found that NS-60YG showed significantly less forward movement than the other two IOLs, even after the intraocular gas had completely disappeared at 1 month postoperatively. Shiraki et al. reported that the IOL position was 0.22 at 1 week after phacovitrectomy with gas tamponade and 0.25 at 1 month afterward [2]. Although they did not state which IOLs they used, these values are very similar to those of XY1 and X-70 in our study. Of interest, the larger optic diameter of X-70 (7.0 mm vs. 6.0 mm for the other two IOLs) did not reduce the anterior shift of IOL. However, in the present study, the circumferential centering of X-70 was better than that of the other IOLs. Therefore, larger optic diameters might contribute to better centering of the IOL in the capsule but not to a reduction of the anterior shift.

We expected that less anterior shift would have minimized the myopic error. However, our analysis did not detect a significant relationship between anterior shift and myopic error. Instead, as reported elsewhere [4], preoperative macular status was found to significantly affect postoperative myopic error. Previous studies also reported that a detached retina at the macula results in an underestimate of the axial length and in myopic shift compared with eyes with macula-on RRD [10]. Our data also suggest that the measurement of axial length needs to be improved, especially in macula-off RRD.

To replicate the endurance of the three IOLs in the eyes with gas tamponade, we performed biomechanical measurements of the force required to move the IOLs anteriorly. In line with a previous study [7], we found that the displacement force of NS-60YG was greater than that of XY1; it was also greater than that of X-70. This result substantiated the better stability of the position of NS-60YG in eyes with gas tamponade. In addition, NS-60YG was characterized by a wider contact angle and lower unit compression load. From these results, we speculate that the haptics of NS-60YG lead to even expansion of the capsule and yield better stability compared with XY1 and X-70.

As found in a previous study [8], our data also suggested that a larger IOL does not necessarily prevent gas-induced myopic shift in phacovitrectomy. Nonetheless, larger IOLs still have benefits during and after surgery in terms of visibility of the entire peripheral retina. Because X-70 is a three-piece IOL, it has the thinnest and narrowest haptics of the three IOLs, as shown in Table 1. Therefore, as also noted elsewhere [7,11], the thickness and width of the haptics are key factors in determining the endurance of IOLs.

This study has some limitations. The sample size was small because the necessary data were missing in several individuals, so they had to be excluded; to elucidate the correlation between postoperative IOL position and myopic error, a further study is required that collects more data than the current study. Second, we did not measure IOL position in phacoemulsification alone, and postoperative IOL position may greatly differ also after this operation. Shiraki et al. showed that the relative IOL position after cataract surgery was 0.36 at 1 month. In the current study, the IOL position of NS-60YG after phacovitrectomy with gas tamponade was 0.35 at 1 month, indicating that implantation of NS-60YG minimizes the unfavorable effect of gas tamponade in phacovitrectomy for RRD. Third, it is unclear whether the anterior shift lasts for a year or more. Last, we included macula-off RRD to collect enough cases for statistical analysis. Macula-off RRD is a critical factor in the postoperative myopic shift of IOL after phacovitrectomy, which may explain why we did not detect a significant effect of each IOL on myopic error after phacovitrectomy with gas tamponade for RRD. To overcome the above-mentioned limitations, a multicenter cohort study is needed to collect more data on macula-on RRD and precisely determine the stability of IOLs.

## 5. Conclusions

In phacovitrectomy for RRD, the IOL NS-60YG shows less anterior shift due to gas tamponade than the IOLs XY1 and X-70. The value of the displacement force might provide useful information on the stability of IOLs after phacovitrectomy for RRD and other retinal diseases that require gas tamponade.

## Figures and Tables

**Figure 1 jcm-11-03438-f001:**
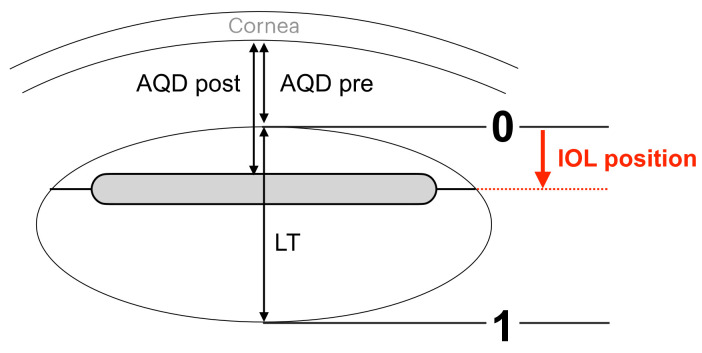
Calculation of intraocular lens position (IOL position) after phacovitrectomy according to the method described by Shiraki et al. [2]. AQD pre, preoperative aqueous depth; AQD post, postoperative aqueous depth; LT, lens thickness; IOL position, intraocular lens position.

**Figure 2 jcm-11-03438-f002:**
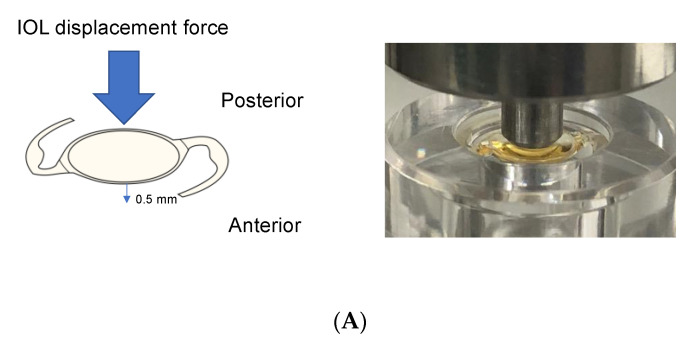
Biomechanical measurements of the intraocular lens (IOL). (**A**) IOL displacement was determined as the force required to move an IOL 0.5 mm forwards. (**B**) Compression load was defined as the force required to decrease the total length of the IOL to 10 mm. (**C**) Contact angle was the total angle of the area where the two haptics were in contact with the wall of the 10 mm diameter holder.

**Figure 3 jcm-11-03438-f003:**
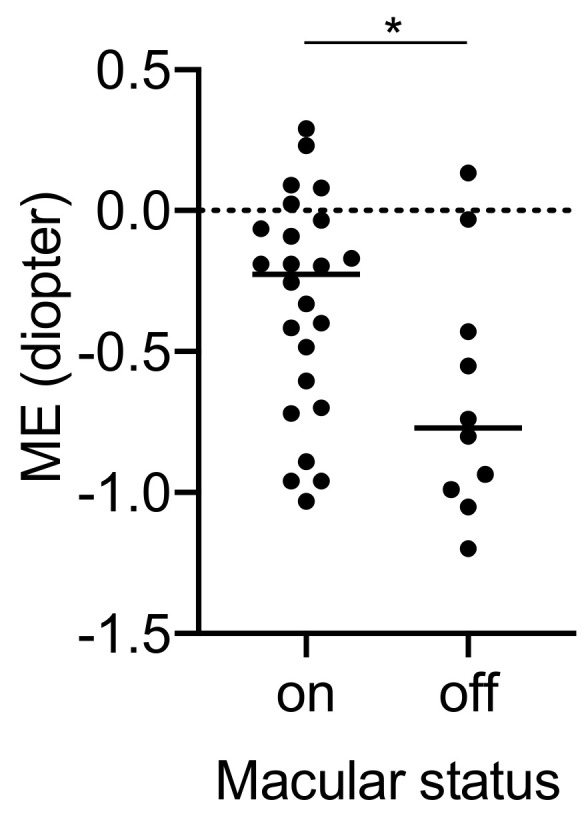
Mean prediction refractive error of intraocular lenses (NS-60YG, XY1, and X-70) between macula-on and -off rhegmatogenous retinal detachment. * *p* < 0.05. ME, mean prediction refractive error.

**Figure 4 jcm-11-03438-f004:**
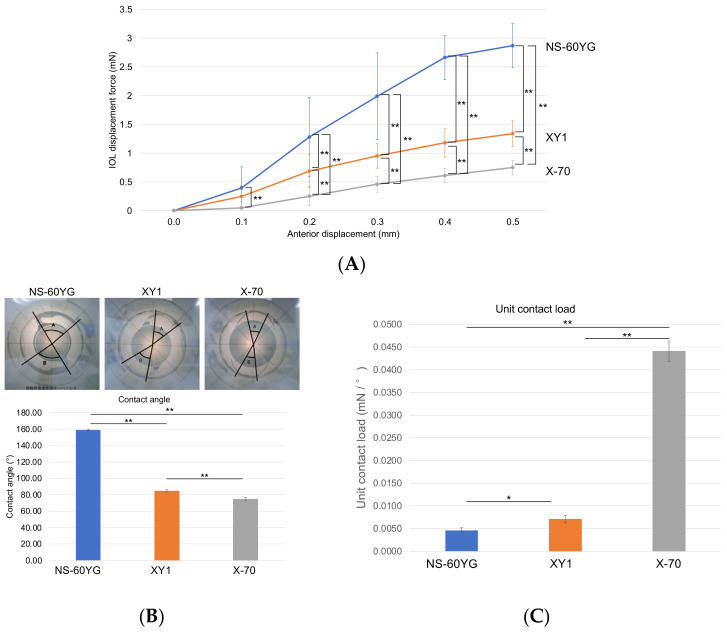
The biomechanical characteristics of the intraocular lenses (IOLs) NS-60YG, XY1, and X-70. (**A**) The IOL displacement force at each 0.1 mm of anterior displacement up to 0.5 mm. Repeated measures two-way analysis of variance was performed. ** *p* < 0.01 (**B**) The comparison of contact angle of the intraocular lenses (IOL). (**C**) Unit contact load (compression load/contact angle). * *p* < 0.05, ** *p* < 0.01. Blue line and bars, NS-60YG; Orange line and bars, XY1; Grey line and bars, X-70.

**Table 1 jcm-11-03438-t001:** Characteristics of intraocular lenses.

	NS-60YG	XY1	X-70
Design	Aspherical, 1 piece	Aspherical, 1 piece	Spherical, 3 piece
Material	Acrylic resin	Acrylic resin	Optic, acrylic resinhaptic, polyvinylidene difluoride
Optic diameter, mm	6.0	6.0	7.0
Total length, mm	13.0	13.0	13.2
Haptics angle, °	0	0	7
Appearance	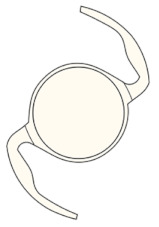	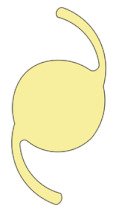	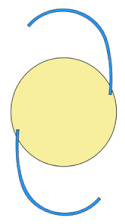

**Table 2 jcm-11-03438-t002:** Characteristics of patients undergoing phacovitrectomy.

Characteristics	Overall	Intraocular Lens Group			*p*-Value
		NS-60YG	XY1	X-70	
Eyes/patients	37/36	15/14	11/11	11/11	-
Age, y	60.6 ± 8.3	60.0 ± 7.4	60.5 ± 8.1	61.6 ± 9.5	0.89
Men/women	26/11	14/1	7/4	5/6	<0.05
Macular status (on/off)	27/10	11/4	8/3	8/3	0.99
Preoperative BCVA, logMAR	0.30 ± 0.60	0.39 ± 0.71	0.21 ± 0.39	0.31 ± 0.61	0.77
Postoperative BCVA, logMAR	−0.02 ± 0.18	0.02 ± 0.24	−0.05 ± 0.12	−0.02 ± 0.10	0.85
Axial length, mm	25.6 ± 1.4	25.7 ± 1.2	25.6 ± 1.2	25.5 ± 1.8	0.97
AQD, mm	2.79 ± 0.33	2.69 ± 0.29	2.91 ± 0.37	2.82 ± 0.28	0.25
ACW, mm	11.69 ± 0.34	11.68 ± 0.40	11.71 ± 0.38	11.66 ± 0.34	0.95
CCT, µm	552.31 ± 37.99	562.86 ± 25.28	546.00 ± 42.77	544.50 ± 43.41	0.43
Lens thickness, mm	4.51 ± 0.28	4.51 ± 0.28	4.51 ± 0.28	4.51 ± 0.28	0.94

All values are mean ± standard deviation unless otherwise indicated. AQD, aqueous depth; ACW, anterior chamber width; BCVA, best-corrected visual acuity; CCT, central corneal thickness; logMAR, logarithm of the minimal angle of resolution.

**Table 3 jcm-11-03438-t003:** Refractive outcomes at 1 month after phacovitrectomy.

	Overall	Intraocular Lens Group			*p*-Value
		NS-60YG (n = 15 Eyes)	XY1 (n = 11 Eyes)	X-70 (n = 11 Eyes)	
MedAE	0.44 ± 0.33	0.31 ± 0.26	0.44 ± 0.36	0.58 ± 0.32	0.14
ME	−0.39 ± 0.39	−0.23 ± 0.34	−0.37 ± 0.44	−0.58 ± 0.32	0.09

All values are mean ± standard deviation. ME, mean refractive prediction error; MedAE, mean median absolute error.

**Table 4 jcm-11-03438-t004:** Parameters of swept source-anterior segment optical coherence tomography and intraocular lens status.

	Overall	Intraocular Lens Group			*p*-Value
		NS-60YG (n = 15 Eyes)	XY1 (n = 11 Eyes)	X-70 (n = 11 Eyes)	
1 week postoperatively					
AQD, mm	4.06 ± 0.33	4.13 ± 0.29	3.99 ± 0.37	4.02 ± 0.28	0.56
IOL position, a.u	0.27 ± 0.07	0.32 ± 0.07	0.24 ± 0.05	0.26 ± 0.04	<0.05
IOL tilt, degree °	4.51 ± 1.95	4.35 ± 2.61	4.61 ± 1.32	4.64 ± 1.37	0.93
IOL decentration, mm	0.26 ± 0.18	0.29 ± 0.18	0.38 ± 0.14	0.13 ± 0.11	<0.01
1 month postoperatively					
AQD, mm	4.19 ± 0.26	4.26 ± 0.26	4.13 ± 0.25	4.13 ± 0.23	0.40
IOL position, a.u	0.30 ± 0.05	0.35 ± 0.05	0.27 ± 0.03	0.28 ± 0.03	<0.01
IOL tilt, degree °	3.94 ± 1.39	3.68 ± 1.74	3.96 ± 1.39	4.37 ± 1.37	0.67
IOL decentration, mm	0.25 ± 0.15	0.27 ± 0.13	0.30 ± 0.17	0.15 ± 0.09	0.13

All values are mean ± standard deviation. AQD, aqueous depth; IOL, intraocular lens; a.u, arbitrary unit.

**Table 5 jcm-11-03438-t005:** Swept source-anterior segment optical coherence tomography parameters and intraocular lens status.

	Mean Prediction Refractive Error	
Variable	*r*	*p*-Value
IOL	−0.037	0.72
Axial length	−0.054	0.35
Preoperative AQD	−0.15	0.64
AQD 1 month postoperatively	0.40	0.35
IOL position 1 month postoperatively	−1.04	0.068
Macular status (on/off)	0.34	0.048

AQD, aqueous depth; IOL, intraocular lens.

## Data Availability

The datasets of this study are available from the corresponding author upon reasonable request.

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
