# Peer review of "Comparison of Postoperative Stability of Intraocular Lenses after Phacovitrectomy for Rhegmatogenous Retinal Detachment"

_jcm, 2022, doi:10.3390/jcm11123438_

Round 1

Reviewer 1 Report

The research article presented on Comparison of postoperative stability of intraocular lenses after phacovitrectomy for rhegmatogenous retinal detachment is interesting and useful for ophthalmic division.

1. Justify for selection of phacovitrectomy, which is combination o phacoemulsification and vitrectomy, however phacoemulsification is the most commonly performed surgery in ophthalmology

2. On page 5 line 148, Why they have taken 37 eyes as there is no correlation to taking 37 eyes from patients.

3. On line 56, author have taking 25 or 27 gauges, is any specific reason to using both gauges, generally small gauges is more preferable.

4. The value of contact angle of NS-60YG i.e. 160° is quite higher in case of intraocular lense (refer https://link.springer.com/content/pdf/10.1007/s40005-017-0323-y.pdf).

5. Is the author have any stability data of the IOL postoperative administration like 1 or 2 years which indicate the linger durability and biocompatibility.

It required extensive revision with some major suggestions.

Author need to rewrite the introduction with the latest literature citation of 2019-2022 years.

Discussion part is very short and required some more,  line 202-250.

Author Response

We thank all the reviewers for providing constructive comments and valuable suggestions for the manuscript. The reviewers’ comments are addressed point-by-point below.

Responses to Reviewer 1

Comment 1: Justify for selection of phacovitrectomy, which is combination of phacoemulsification and vitrectomy, however phacoemulsification is the most commonly performed surgery in ophthalmology.

Response 1: Thank you so much for your comment. We started this study because we had been noticing a difference in IOL stability during and after gas tamponade. In addition, we had been wondering whether the 7 mm diameter of a three-piece IOL provides better IOL stability in the anterior chamber after gas tamponade. Therefore, we limited participants to patients with rhegmatogenous retinal detachment. We did not compare IOL stability in phacoemulsification alone, so we cannot exclude the possibility that IOL position differed significantly even without gas tamponade. To address this concern, we have added the following sentences to the limitations described on p. 9, line 256: “Second, we did not measure IOL position in phacoemulsification alone, and postoperative IOL position may greatly differ also after this operation. Shiraki et al. showed that the relative IOL position after cataract surgery was 0.36 at 1 month. In the current study, the IOL position of NS-60YG after phacovitrectomy with gas tamponade was 0.35 at 1 month, indicating that implantation of NS-60YG minimizes the unfavorable effect of gas tamponade in phacovitrectomy for RRD.”

Comment 2: On page 5 line 148, Why they have taken 37 eyes as there is no correlation to taking 37 eyes from patients.

Response 2: Thank you for this comment. This was a retrospective study, so all eligible patients were included who were treated during the study period (May 2020 to March 2021). We believe that it is clear from “37 eyes in 36 patients” that 2 eyes were included in 1 patient and 1 eye in all other patients and do not think that we need to explicitly state this fact. To make the numbers of eyes and patients clearer, we have added the following sentence on p. 5, line 154: “The study included data from 37 eyes in 36 patients.”

Comment 3: On line 56, author have taking 25 or 27 gauges, is any specific reason to using both gauges, generally small gauges is more preferable.

Response 3: Thank you so much for your question. In the current study, three surgeons performed phacovitrectomy for RRD. Because this was a retrospective study, the gauge was not standardized, and each surgeon used the gauge they preferred. We have added the following sentence on p. 2, line 63: “The decision to use 25 or 27 gauge was made by the surgeon.”

Comment 4: The value of contact angle of NS-60YG i.e. 160° is quite higher in case of intraocular lens (refer https://link.springer.com/content/pdf/10.1007/s40005-017- 0323-y.pdf).

Response 4: Thank you so much for your comment and the reference. Actually, while preparing our original manuscript, we tried to find publications on the contact angle of haptics, but we were unable to find any suitable ones. We read the paper that you recommended, and it seems to use contact angle in a different way than we did. If we understand the article correctly, the authors evaluated the angle of water droplets on the surface of IOLs to compare differences in their surface properties, which is a different approach to the one used in our study. Therefore, we do not think this reference is suitable for inclusion in our manuscript.

Comment 5: Does the author has any stability data of the IOL postoperative administration like 1 or 2 years which indicate the linger durability and biocompatibility.

Response 5: Thank you so much for your question. We were asking ourselves the same thing. Unfortunately, we do not have any data on postoperative IOL stability over the course of 1 or 2 years. We have added the following sentence to the limitations on p. 9, line 261: “Third, it is unclear whether the anterior shift lasts for a year or more.”

Comment 6: It required extensive revision with some major suggestions. Author needs to rewrite the introduction with the latest literature citation of 2019-2022 years. Discussion part is very short and required some more, line 202-250.

Response 6: Thank you so much for your suggestion. We have added sentences and references to the Introduction and Discussion (p. 1, line 42 to p. 2, line 47 and p. 9, lines 256-268).

Reviewer 2 Report

In the study ‘Comparison of postoperative stability of intraocular lenses after phacovitrectomy for rhegmatogenous retinal detachment’ by Akiyama et al. the postoperative refraction error of three different intraocular lenses is investigated.

This is an exciting question, but it is being investigated with a very small and retrospective study. The method is unclear. Why were not only patients with macula-on included? It would have been much more reliable to determine the postoperative refractive error. In addition, it remains unclear why results from the laboratory have to be mentioned again. They are not given in the abstract. Which parameters are really decisive?

The work must be described much more clearly so that it is understandable.

Abstract

·      Page 1, line 21: what do the numbers mean? This is unclear and needs to be better explained.

·      Page 3&4: This is a retrospective study. Why is a laboratory result being brought in now? Nothing is mentioned about this in the abstract. What is the reason?

·      Page 5, table 2: the number of patients operated on is very small for a retrospective study, moreover, in some patients the macula was detached. How can the lens power then be calculated correctly? How useful is the parameter postoperative refraction?

·      Table 3: the postoperative refractive error is not a meaningful parameter in patients with macula-off retinal detachment.

·      Table 4: we learn from this table that the values in the abstract are 'IOL position, a.u', although it remains unclear what exactly this is supposed to mean. Is this really the most important parameter?

Author Response

We thank all the reviewers for providing constructive comments and valuable suggestions for the manuscript. The reviewers’ comments are addressed point-by-point below.

Responses to Reviewer 2

Comment 1: In the study ‘Comparison of postoperative stability of intraocular lenses after phacovitrectomy for rhegmatogenous retinal detachment’ by Akiyama et al. the postoperative refraction error of three different intraocular lenses is investigated. This is an exciting question, but it is being investigated with a very small and retrospective study. The method is unclear. Why were not only patients with macula-on included? It would have been much more reliable to determine the postoperative refractive error. In addition, it remains unclear why results from the laboratory have to be mentioned again. They are not given in the abstract. Which parameters are really decisive? The work must be described much more clearly so that it is understandable.

Response 1: We totally agree with your opinion. In a clinical study, the number of cases is critical for obtaining reliable results. However, in the present study, the choice of IOL depended completely on each surgeon’s preference. The three IOLs were used in our hospital for a relatively short time, which limited the number of participants. In addition, the necessary data were missing in some cases. We mention this point as a limitation on p. 9, lines 253-256.

To make the method more understandable, we have modified Figure 1 to explain how relative IOL position was measured and added the following sentence on p. 1, lines 18-20: “IOL position was determined as follows: IOL position = (postoperative anterior chamber depth [ACD] – preoperative ACD) / lens thickness.”

Regarding macula-on or -off RRD, the sample size would have been even smaller if we had studied only one or the other, so we included both types. We have added a sentence about this topic to the limitations. As regards the laboratory data, please see our response to Comment 3 below.

Comment 2: Page1, line 21: what do the numbers mean? This is unclear and needs to be better explained.

Response 2: Thank you so much for your question. These numbers represent the IOL position. However, the Abstract did not contain any information on IOL position. We have added the explanation of how IOL position was calculated to the Abstract, as follows (p. 1, lines 18-20): “IOL position was determined as follows: IOL position = (postoperative anterior chamber depth [ACD] – preoperative ACD) / lens thickness.”

Comment 3: Page 3&4: This is a retrospective study. Why is a laboratory result being brought in now? Nothing is mentioned about this in the abstract. What is the reason?

Response 3: Thank you so much for your questions. In contrast to other studies, the present study compared two single-piece and one three-piece IOL. To our knowledge, no data are available on the biomechanical measurement of X-70, which is why we performed biomechanical measurements. We have added the following sentence to the Abstract (p. 1, line 23): “To replicate the anterior shift of IOL position ex vivo...”

Comment 4: Page 5, table 2: the number of patients operated on is very small for a retrospective study, moreover, in some patients the macula was detached. How can the lens power then be calculated correctly? How useful is the parameter postoperative refraction?

Response 4: We totally agree with your opinion. The choice of IOL was completely up to each surgeon. As mentioned in our response to Comment 1, the number of cases is critical for obtaining reliable results in a clinical study. The number of participants in our study was small because the choice of IOL was up to each surgeon, and the three IOLs were used in our hospital for a relatively short period. In addition, the necessary data were missing in some cases. We now mention this point as a limitation on page 9, lines 253-256.

              Regarding the determination of IOL in the case of macula-off RRD, applanation A-scan ultrasonography was used. We have revised the respective sentence on p. 3, lines 99-101, as follows: “Axial lengths were measured by optical biometry (IOL Master; Carl Zeiss) in case of macula-on RRD or applanation A-scan ultrasonography (Tomey Corp, Nagoya, Japan) in case of macula-off RRD.”

Comment 5: Table 3: the postoperative refractive error is not a meaningful parameter in patients with macula-off retinal detachment.

Response 5: Thank you so much for your comment. Because macula-off RRD is a critical factor in postoperative error myopic shift, we should have performed the statistical analysis with the data from macula-on RRD. However, anterior shift due to gas tamponade was the main topic of interest in the present study. We performed this analysis because anterior shift is also known to cause myopic shift. We have added the following sentence to the limitations on p. 9, lines 263-268: “Last, we included macula-off RRD to collect enough cases for statistical analysis. Macula-off RRD is a critical factor in postoperative myopic shift of IOL after phacovitrectomy, which may explain why we did not detect a significant effect of each IOL on myopic error after phacovitrectomy with gas tamponade for RRD. To overcome the above-mentioned limitations, a multicenter cohort study is needed to collect more data on macula-on RRD and precisely determine the stability of IOLs.”

Comment 6: Table 4: we learn from this table that the values in the abstract are 'IOL position, a.u', although it remains unclear what exactly this is supposed to mean. Is this really the most important parameter?

Response 6: Thank you for your question. We have added the information on how to calculate IOL position to the Abstract, as follows (p. 1, lines 18-20): “IOL position was determined as follows: IOL position = (postoperative anterior chamber depth [ACD] – preoperative ACD) / lens thickness.”

              We started this study because we had been noticing a difference in IOL stability during and after gas tamponade. In addition, we were wondering whether the 7 mm diameter of the three-piece IOL provides better IOL stability in the anterior chamber after gas tamponade. IOL position determined by CASIA2 parameters is not greatly affected by the axial length compared with the postoperative ACD. In fact, our analysis could not detect any difference in postoperative ACD between IOLs. We believe that IOL position relative to lens thickness is very useful for comparing postoperative IOL position in individuals with different axial lengths.

Round 2

Reviewer 2 Report

All remarks have been answered